# A Robust Fermentation Process for Natural Chocolate-like Flavor Production with *Mycetinis scorodonius*

**DOI:** 10.3390/molecules27082503

**Published:** 2022-04-13

**Authors:** Marina Rigling, Fabienne Heger, Maria Graule, Zhibin Liu, Chen Zhang, Li Ni, Yanyan Zhang

**Affiliations:** 1Department of Flavor Chemistry, Institute of Food Science and Biotechnology, University of Hohenheim, Fruwirthstraße 12, 70599 Stuttgart, Germany; marina.rigling@uni-hohenheim.de (M.R.); fabienne-heger@gmx.de (F.H.); maria.graule@gmx.de (M.G.); 2Institute of Food Science and Technology, College of Biological Science and Engineering, Fuzhou University, Fuzhou 350108, China; liuzhibin@fzu.edu.cn (Z.L.); zhangchenfj@sina.com (C.Z.); nili@fzu.edu.cn (L.N.)

**Keywords:** green tea, chocolate-like flavor, fermentation process, *Mycetinis scorodonius*, basidiomycetes, bioreactor

## Abstract

Submerged fermentation of green tea with the basidiomycete *Mycetinis scorodonius* resulted in a pleasant chocolate-like and malty aroma, which could be a promising chocolate flavor alternative to current synthetic aroma mixtures in demand of consumer preferences towards healthy natural and ‘clean label’ ingredients. To understand the sensorial molecular base on the chocolate-like aroma formation, key aroma compounds of the fermented green tea were elucidated using a direct immersion stir bar sorptive extraction combined with gas chromatography–mass spectrometry–olfactometry (DI-SBSE-GC-MS-O) followed by semi-quantification with internal standard. Fifteen key aroma compounds were determined, the most important of which were dihydroactinidiolide (odor activity value OAV 345), isovaleraldehyde (OAV 79), and coumarin (OAV 24), which were also confirmed by a recombination study. Furthermore, effects of the fermentation parameters (medium volume, light protection, agitation rate, pH, temperature, and aeration) on the aroma profile were investigated in a lab-scale bioreactor at batch fermentation. Variation of the fermentation parameters resulted in similar sensory perception of the broth, where up-scaling in volume evoked longer growth cycles and aeration significantly boosted the concentrations yet added a green note to the overall flavor impression. All findings prove the robustness of the established fermentation process with *M. scorodonius* for natural chocolate-like flavor production.

## 1. Introduction

Many natural flavors are commonly extracted from their natural plant sources, which lead to high prices due to limited accessibility and natural drawbacks such as crop failure, as well as time-consuming processing. Nevertheless, consumers explicitly prefer natural food ingredients over synthesized ones. A lot of research was documented to produce natural food ingredients via biotechnological processes, among which bio-flavors have been investigated intensively attributed to their low costs and natural labelling in the European Union (EU) for food application [1]. Biotechnological flavor synthesis, i.e., fermentation (*de novo*) and bioconversion, is preferred compared with the highly time-consuming and expensive processing of plant flavorings. More importantly, the EU legislation on flavors (EEC No. 1334/2008) defines that “natural flavoring substances shall mean a flavoring obtained by appropriate physical, enzymatic or microbiological origin either in the raw state or after processing for human consumption by one or more of the traditional food preparation processes”, which indicates that natural flavors are obtained by microbiological processes using natural materials as substrates. Accordingly, natural flavors biosynthesized by basidiomycetes offer an alternative to those extracted from natural plant sources [2], since the odor spectra produced by basidiomycetes is most like that of plants [1,3]. Basidiomycetes, as the highest developed class of fungi, have shown their impressive capability to produce numerous natural flavors from a series of different plant-based substrates in the last years [1,4,5,6,7]. Their unique and complex sets of extracellular enzymes [8,9] allow basidiomycetes to adapt well to not only different plant biopolymers but also simple nutrients such as sugars or amino acids to synthesize flavorings *de novo* [1,2]. It has been further demonstrated that these flavoring substances are identical to compounds extracted from plants. The basidiomycete *Mycetinis scorodonius*, known as garlic parachute, is a wild grown mushroom with distinct garlic flavor, which makes it favorable as a seasoning mushroom in the kitchen. Apart from that, the fungal mycelium is of high interest in the current research. A series of peroxidases have been identified in *M. scorodonius*, which makes it interesting for biotechnological applications [10,11]. In this research, *M. scorodonius* was found as a promising candidate for natural flavor generation after fermentation of green tea infusion. Within 64 h, the fermented green tea exhibited a pleasant chocolate-like and malty flavor impression. The pleasant aroma profile was qualitatively, quantitatively, and organoleptically investigated by means of a direct immersion stir bar sorptive extraction combined with gas chromatography–mass spectrometry–olfactometry (DI-SBSE-GC-MS-O). To check the robustness of the fermentation process, we transferred the fermentation process to a lab-scale bioreactor and the influence of varied fermentation parameters (medium volume, light protection, agitation rate, pH, fermentation temperature, and aeration) on the chocolate-like flavor biosynthesis was tested (Figure 1).

## 2. Results and Discussion

### 2.1. Qualitative, Quantitative, and Organoleptic Investigation on Chocolate-like Aroma Formed by the Basidiomycete Mycetinis scorodonius

In our previous study, natural aromatization of non-alcoholic drinks such as green tea [7,12] and alcohol-free beer from wort [4] have been implemented via basidiomycetes biocatalysis. It was described that the basidiomycete *Flammulina velutipes* has the capability to tune the green tea flavor into a chocolate-like and nutty flavor after a submerged fermentation processing [7]. Currently, we found that the basidiomycete *Mycetinis scorodonius* was also able to shift the original green floral and soapy flavor of the green tea infusion into a pleasant chocolate-like and malty flavor after 64 h (Table 1), which makes it another potential candidate for natural chocolate-like flavor generation. Concurrently, the color of the green tea broth changed during fermentation from originally yellowish and greenish to dark brown, which is due to oxidation and polymerization of tea catechins (data not shown). To the best of our knowledge, no research on the application of *M. scorodonius* for natural flavor generation is available to date. Nevertheless, a series of enzymes, e.g., dye-decolorizing peroxidases (DyP-type) [10,11] or carotenoid-cleaving enzymes (MsP1 and MsP2) [13], have already been discovered and decoded from the fungus so far. Since tea is a natural source of carotenoids, the carotenoid-cleaving enzymes are especially likely to play an important role in the flavor change during fermentation.

After isolation of the volatiles by means of a direct immersion stir bar sorptive extraction (DI-SBSE), the odorants of the fermented broth were analyzed by gas chromatography–mass spectrometry–olfactometry (GC-MS-O) combined with a thermal desorption unit (TDU) and cooled injection system (CIS). In total, 100 odor-active regions were perceived by three trained assessors (Table 2), correspondingly 80 of which were identified based on retention index, odor attributes, and authentic standards if available. Despite identified typical green tea volatiles imparting green or floral odor impressions such as hexanal, (*Z*)-3-hexen-1-ol, and (*E*)-2-octen-1-al, a series of volatiles with spicy, chocolate-like, or caramel-like odor impressions such as 2-ethyl-3,5(6)-dimethylpyrazine, isovaleraldehyde, and coumarin, already identified in chocolate or related products [14,15,16], were found in the fermented broth. All odorants assigned were semi-quantified by internal standard (Table 2).

Calculation of their odor activity values (OAVs) based on the corresponding odor threshold in water [17,18,19,20,21,22,23], revealed only 15 compounds (OAV ≥ 1), namely isovaleraldehyde, octanal, (*Z*)-3-hexen-1-ol, nonanal, 2-ethyl-3,5(6)-dimethylpyrazine, linalool, acetophenone, (*S*)-verbenone, (*Z*)-geraniol, (*Z*)-carveol, (*Z*)-jasmone, 2-methoxy-4-vinylphenol, dihydroactinidiolide, and coumarin to be important for the overall chocolate-like and malty aroma (Table 2). Dihydroactinidiolide (woody and musk-like odor impression, OAV 345), isovaleraldehyde (nutty and chocolate-like odor impression, OAV 79), and coumarin (amaretto-like and spicy odor impression, OAV 24) seem to be the main driving forces for the formation of the chocolate-like flavor impression. Dihydroactinidiolide is also identified as an important aroma compound in various teas [24]. Dihydroactinidiolide, a C_11_-terpenic lactone, is a carotenoid degradation product via enzymatic *β*-ionone pathway or thermal degradation [25,26]. Fungal peroxidases for cleavage of *β*-carotene towards *β*-ionone have been reported in the basidiomycetes *M. scorodonius* and *Lepista irina* [27,28]. Isovaleraldehyde is commonly described in a series of food, e.g., coffee, tea, or potatoes [29,30,31]; moreover, it is defined as one of the most important aroma compounds in cocoa and chocolate [32]. As a Strecker aldehyde, isovaleraldehyde is a product of the amino acid degradation, especially leucine [33]. Similar enzymatic biosynthesis pathways towards isovaleraldehyde or phenylacetaldehyde are well documented in basidiomycetes [34]. Coumarin is a natural occurring lactone. However, little is known so far about synthesis in microbes. However, a series of coumarin derivates have been reported in different basidiomycetes [35]. Phenylalanine ammonia lyase and *β*-glycosidase were considered as key enzymes in the coumarin derivates biosynthesis from glucose [36]. In comparison with the chocolate-like flavor produced from *F. velutipes* [7] that imparted a nutty note besides the chocolate-like one, the elucidated aroma profile showed that a chocolate-like flavor can be obtained from different combinations of odorants. Only 2-ethyl-3,5-dimethylpyrazine was found as a key aroma compound in both fermented green tea infusions obtained from *F. velutipes* and *M. scorodonius*. The chocolate-like flavor formed by *F. velutipes* was mainly attributed to a mixture of methyl jasmonate (OAV 4062), *β*-ionone (OAV 896), and *β*-damasceone (OAV 79). Nevertheless, typical cocoa or chocolate odorants such as furaneol, phenylacetic acid, and skatole also contributed to the generation of chocolate-like flavor [7].
molecules-27-02503-t002_Table 2Table 2Odor-active compounds identified in the fermented broth by means of DI-SBSE-GC-MS-O.No.RICompoundOdor ImpressionConcentration (µg/L)Odor Threshold (µg/L)OAVIdentification19162-methylbutanalchocolate-like, nutty, fermented10.17 ± 0.8122.5 ^§^<1RI, odor, MS2920isovaleraldehydenutty, chocolate-like, floral15.83 ± 0.700.279RI, odor, MS3944ethanolethereal, alcoholic3.21 ± 0.0110,000<1RI, odor, MS4983valeraldehydefruity, nutty, chocolate-like- *

RI, odor510302-butanolsweetish, fruity17.22 ± 2.92500<1RI, odor, MS61038unknownfresh, floral----71050unknownfermented, brown, hay-like, malty----81078hexanalfresh, green, grassy2.58 ± 0.424.5<1RI, odor, MS91096unknownchocolate-like, caramel-like-

-101115isoamyl acetatesweetish, banana-like, fruity23.22 ± 2.2526<1RI, odor, MS111177heptanalfatty, rancid15.01 ± 2.9816<1RI, odor, MS131185limonenefruity, citrus-like, herbal0.69 ± 0.0910<1RI, odor, MS141199eucalyptolherbal, medicinal, ethereal0.33 ± 0.051.3<1RI, odor, MS151207isoamyl alcoholalcoholic, fruity14.72 ± 1.91250<1RI, odor, MS161232ethyl hexanoatewine-like, fruity- *--RI, odor171239unknownbrown, floral, hay-like----181251amyl alcoholfermented, yeast-like, bread-like- *--RI, odor191254pentanolsweetish, alcoholic11.51 ± 0.734000<1RI, odor, MS2012622-methylpyrazinebrown, musty, earthy10.79 ± 2.0460<1RI, odor, MS211279octanalfruity, floral0.73 ± 0.140.71RI, odor, MS221293hydroxy acetonepungent, ethereal0.23 ± 0.0510 ^§^<1RI, odor, MS231298unknownhay-like, green----241308(*Z*)-3-hexenyl acetatefruity, green, pear-like- *--RI, odor2513172,5-dimethylpyrazinepeanut-like, nutty, roasted25.19 ± 3.83800<1RI, odor, MS2613252,6-dimethylpyrazineroasted, coffee-like2.33 ± 0.11400<1RI, odor, MS2713292-ethylpyrazinepeanut-like, chocolate-like1.62 ± 0.216000<1RI, odor, MS2813316-methyl-5-hepten-2-onemusty, brown5.57 ± 0.6450<1RI, odor, MS291350unknownsharp, pungent----301351hexanolfruity, alcoholic, ethereal7.71 ± 0.662500<1RI, odor, MS311374unknownfresh, fruity, citrus-like----3213762-ethylhexyl acetateherbal, hay-like, sweetish- *--RI, odor331380(*Z*)-3-hexen-1-olgreen, leafy, citrus-like157.72 ± 14.89394RI, odor, MS341385nonanalfatty, waxy, brown2.70 ± 0.6513RI, odor, MS3513862-ethyl-5-methylpyrazinechocolate-like, coffee-like-*--RI, odor3613982,3,5-trimethylpyrazineroasted, chocolate-like, nutty5.00 ± 0.66400<1RI, odor, MS371428(*E*)-2-octen-1-alwaxy, fatty, green- *--RI, odor381445(*Z*)-linalool oxidewoody, terpenic- *--RI, odor3914442-ethyl-3,6-dimethylpyrazinecoffee-like, chocolate-like3.27 ± 0.320.48RI, odor, MS401444acetic acidsourish, vinegar-like0.02 ± 0.012000<1RI, odor, MS4114511-octen-3-olearthy, licorice1.12 ± 0.032<1RI, odor, MS421451unknownhay-like----4314572-ethyl-3,5-dimethylpyrazinecocoa-like, musty, roasted1.00 ± 0.1911RI, odor, MS4414576-methylhept-5-en-2-olsweetish, green- *--RI, odor451466(*E*)-linalool oxide (furanoid)woody, terpenic, green- *--RI, odor4614832-ethyl-1-hexanolfresh, fruity, sweetish15.62 ± 1.47270,000<1RI, odor, MS471488decanalherbal, fresh0.93 ± 0.192<1RI, odor, MS481509benzaldehydebitter almond-like77.13 ± 3.71350<1RI, odor, MS491539linaloolfloral, sweetish, fruity1.45 ± 0.210.1410RI, odor, MS501551octyl formatefresh, aromatic- *--RI, odor511589lilac aldehyde Dfresh, floral- *--RI, odor521593terpinen-4-olspicy, terpenic50.11 ± 4.91150<1RI, odor, MS531597unknownbrown, burnt, roasted----541606(*E*)-2-octen-1-olgreen, vegetable-like- *--RI, odor551615γ-butyrolactonefaint, sweetish, rancid1.94 ± 0.385<1RI, odor, MS561628phenylacetaldehydefloral2.37 ± 0.134<1RI, odor, MS571638acetophenonebrown, musty, earthy115.52 ± 11.13652RI, odor, MS5816452,6-dimethyl-5-hepten-1-oloily, waxy, musty- *--RI, odor591688*α*-terpineolethereal, terpenic31.93 ± 2.85330<1RI, odor, MS6016923-methyl-2(5H)-furanonechocolate-like, woody,67.10 ± 8.1589<1RI, odor, MS611698(*S*)-verbenonespicy, menthol-like22.34 ± 4.012.59 ^§^2RI, odor, MS621716dihydrocarveolgreen, mint-like2.11 ± 0.627.5 ^§^<1RI, odor, MS631720citralcitrus-like0.49 ± 0.0832<1RI, odor, MS641725(*E*)-linalool oxide (pyranoid)terpenic, floral- *--RI, odor651754(*E*)-linalool 3,7-oxidehay-like, woody- *--RI, odor661763methyl salicylateminty, sweetish, floral21.01 ± 2.4940<1RI, odor, MS671790(*Z*)-geraniolsweetish, citrus-like, floral, fruity10.27 ± 1.143.23RI, odor, MS681807unknownfermented, malty, chocolate-like----691825(*Z*)-carveolherbal, spearmint-like, sweetish25.87 ± 2.4646RI, odor, MS701831hexanoic acidcheesy, fatty66.44 ± 2.213000<1RI, odor, MS711837unknownfruity, citrus-like, caramel-like----721880geranyl isovalerategreen, apple-like- *--RI, odor731898phenylethanolethereal, fresh, rose-like10.73 ± 1.6286<1RI, odor, MS741909unknownearthy, fungal----751934(*Z*)-jasmonefloral, fruity, woody2.94 ± 0.391.92RI, odor, MS761956unknowncotton candy-like, sweetish----771977*β*-ionon-5,6-epoxidefruity, berry-like- *--RI, odor782014unknownchocolate-like, brown, woody----792025unknownamaretto-like, caramel-like----802049octanoic acidrancid, fatty, vegetable-like, green147.62 ± 5.613000<1RI, odor, MS812085unknowncherry-like, musty, woody----822106unknownethereal, musty----842124unknowncitrus-like, vanilla-like, floral----852156nonanoic acidfatty, coconut-like, waxy81.11 ± 3.913000<1RI, odor, MS8621782-methoxy-4-vinylphenolroasted, broth-like71.11 ± 10.45194RI, odor, MS872184unknownbrown, hay-like----882222methyl anthranilatefruity2.17 ± 0.867.73<1RI, odor, MS892231*α*-cadinolbrown, woody- *--RI, odor902263decanoic acidfermented, rancid, sourish, fruity71.93 ± 16.061000<1RI, odor, MS912320methyl jasmonatefloral, jasmine-like, citrus-like- *--RI, odor922332dihydroactinidiolidewoody, musk-like, fermented638.04 ± 93.051.85 ^§^345RI, odor, MS932362unknownfresh, vanilla-like, citrus-like----942371coumaranbrown, cinnamon-like, spicy-*--RI, odor952421unknownfruity, sourish----962432coumarinamaretto-like, spicy815.95 ± 70.293424RI, odor, MS972472skatolefloral, pungent, stable-like0.74 ± 0.042<1RI, odor, MS982474dodecanoic acidmusty, fatty, sourish26.27 ± 2.8110,000<1RI, odor, MS9924965-hydroxymethylfurfuralchamomile-like, waxy- *--RI, odor1002547vanillinvanilla-like, creamy, sweetish30.62 ± 2.9535<1RI, odor, MS* no authentic commercial standard available. ^§^ determined after [19]. RI—retention index after Kovats. MS—mass spectrum.


Five sensory descriptors, namely chocolate-like, sweetish, malty, caramel-like, and woody were defined as the representative aroma attributes of the fermented green tea broth by the three trained assessors. The intensities of the given odor attributes were rated and compared between the fermented broth and the corresponding aroma model by a panel (*n* = 12). The result of the aroma recombination study demonstrated that the characteristic aroma of the fermented green tea broth with *M. scorodonius* can be simulated well by combining fifteen key odorants in their respective concentrations in water (Figure 2). The performed one-way analysis of variance (*p* > 0.05) postulated no significant difference between the fermented green tea broth and the reconstituted aroma model.

### 2.2. Robustness of Bioreactor Process for Natural Chocolate-like Flavor Production

Brown flavors, e.g., chocolate flavors, have a high global market share. To satisfy explicit consumer demand towards natural food ingredients and implementation of industries, reliable and robust biotechnological processes for production of natural flavors are indispensable. To gain more knowledge on our developed biotechnological process for chocolate-like flavor production, the process was transferred to a lab-scale bioreactor (*V_max_* 4.5 L) after the successful fermentation was set up in shake flasks. Several fermentation parameters (medium volume, light protection, agitation rate, temperature, pH, and aeration) were altered to check the robustness of the fermentation processing on the production of chocolate-like flavors. In the first steps, medium volume was increased to 1 L and 2.5 L, respectively, while the basic fermentation parameters were kept stable (24 °C, 150 rpm). We found that a higher fermentation volume evoked longer growth cycles to form the similar flavor impression of the broth in comparison to the shake flask fermentation broth (250 mL/500 mL); the fermentation times were necessarily prolonged to 68 h for 1 L and 70 h for 2.5 L, respectively (Table 1). Semi-quantification of the 15 key aroma compounds revealed differences in concentration for some compounds, e.g., (*Z*)-3-hexen-1-ol and linalool (Table 3), which might be due to the time lag to obtain the desired aroma profile. Nevertheless, the overall sensory impressions and intensities of the broth obtained in flask fermentation as well as the two broths from the bioreactor with different medium volumes were quite similar. pH Development and partial dissolved oxygen (*p*O_2_) were monitored in the bioreactor fermentation. It was shown that the pH dropped strongly in the first hours of fermentation (<12 h) but stabilized around pH 4 afterward in both 1 L and 2.5 L batches, respectively. The pH drop might be attributed to CO_2_ and organic acid metabolites from the mycelia. The slight increase in pH at the end of the fermentation might be attributed to dying cells and therefore less production of CO_2_ and acid metabolites [37]. We also found differences in the development of dissolved oxygen in the fermentation broth. While *p*O_2_ in the 2.5 L batch decreased rapidly from 100% to 10% in the first 20 h and kept low, the decrease in the 1 L batch was less rapid and increased again to almost 40% between 20 h and 50 h (Figure 3). For the next approaches, the 1 L batch fermentation in the bioreactor was set as the reference in regard to flavor perception and aroma profile.

Then, a comparison of fermentation in darkness (fermentation vessel covered with aluminum foil) vs. light (normal day light with night cycle) with 1 L medium volume was conducted. It is known that light can have an influence on the metabolism of basidiomycetes regarding secondary metabolites [38,39]. For the production of fruiting bodies, light is even essential to induce fructification [40]. However, the development of *p*O_2_ and pH between light and dark fermentation was similar (Figure 3). A small influence on the sensory perception of the fermented broths was detected; the malty side note was not perceived from the fermented broth in darkness (Table 1). Therefore, all further experiments were conducted with an uncovered fermentation vessel.

When adjusting the agitation rate to 100 rpm or 250 rpm, a similar sensory profile was obtained compared with the standard 1 L batch (150 rpm). All batches produced a comparable fermentation broth with chocolate-like and malty/licorice-like aroma impression, despite some differences in the concentrations of the key aroma compounds that were elucidated (Table 1 and Table 3). A closer look at *p*O_2_ and pH revealed that similar behavior was detected in the pH development, whereas the *p*O_2_ was constantly higher over fermentation time for the batch with 250 rpm (Figure 3). Higher speed might increase the metabolites, since higher speed not only leads to higher oxygen intake from the headspace but also increases shear stress on the mycelia, resulting in smaller particle sizes. We found the formation of the smaller mycelia particle sizes under 250 rpm (data not shown), but there was no obvious influence of the perceived chocolate-like flavor impression. Nevertheless, significant higher concentrations of (*Z*)-carveol and coumarin as well as significant lower concentrations of 2-ethyl-3,5-dimethylpyrazine and nonanal were measured in comparison with the standard bioreactor batch with 150 rpm (Table 3).

Temperature robustness of the fermentation process was conducted by adjusting temperatures to 18 °C and 30 °C, respectively. No obvious change in the mycelia appearance (size, distribution) was observed with varying temperatures (data not shown). At 18 °C the *p*O_2_ showed a slower decrease over fermentation time compared with the standard 1 L batch, which might result from the lower metabolism of *M. scorodonius*. At 30 °C, an increase of *p*O_2_ up to 60% was observed after an initial decrease to 20%. For pH, no differences were observed (Figure 3). Changes of temperature showed effects on the concentrations of specific aroma compounds, e.g., isovaleraldehyde, 2-ethyl-3,5(6)dimethylpyrazine (significant increase) (Table 3); nevertheless, the overall aroma perception and intensity of the standard bioreactor batch and the two broths from different temperatures were comparable (Table 1). Variation occurred probably foremost due to physiological changes in the independent biological batches and secondly because of individual optima of the mycelia and secreted enzymes in terms of temperature.

In this study, phosphate buffers (potassium dihydrogenphosphate and disodium phosphate) were used for pH adjustment. For this, the tea infusion was prepared directly in a buffer solution instead of water. In the non-buffered fermentation, it was found that the original pH of the green tea infusion (pH 5.3) decreased to pH 4 in the first 12 h and stabilized there until 50 h fermentation time. Afterwards, the pH increased slightly to 4.5 towards the end of fermentation. In the buffered fermentations, the pH was stable at the selected point and only small decreases (up to 0.1 pH) after 60 h fermentation were observed. Immense differences in dissolved oxygen proportion were found between the buffered fermentations and the standard fermentation, which might be relevant to different physiological activities at the selected pH (Figure 3). Semi-quantification revealed the pH 5 batch to be more similar in the aroma profile than the pH 8 batch (Table 3). The pH variation probably affected the individual optima of the mycelia and secreted enzymes. It has been reported that enzymes secreted by basidiomycetes were more active in a slightly sour environment than in an alkaline one [41]. In sensory evaluation, the batch at pH 8 clearly showed sensory influence from the used salt buffer. Although both batches showed good accordance in the sensory to the non-buffered batch, the non-buffered fermentation imparted with the most balanced flavor perception and with the highest intensity (Table 1). In the follow-up studies, the fermentations should be repeated by regulating the pH with sodium hydroxide or phosphoric acid throughout the fermentation instead of using salt buffers.

Finally, active aeration with pressured air was conducted in the fermentation processing. Due to the high amount of saponins and other polyphenols in the tea infusion, the active aeration provoked strong foaming of the broth; therefore, anti-foam agent (Antifoam 204, Sigma-Aldrich) was needed to be added to the fermentation broth to suppress excessive foaming. Aeration was set to 100% of original O_2_ proportion. We observed that in the first 20 h a high gas flow (up to 2 L/min) was needed to keep a stable *p*O_2_, while the gas flow leveled around 1 L/min was enough between 20 h and 68 h. Aeration showed no effect on pH development in comparison with the standard and non-aerated batches (Figure 3). A big difference was observed in the overall odor profile of the aerated fermentation broth; an extra green odor note was perceived besides the desired chocolate-like impression (Table 1). Compared with the standard non-aerated bioreactor fermentation, the aroma profile of this approach showed the most differences in semi-quantification. All concentrations of the 15 key odorants were significantly higher (up to factor 10) than those in the other fermentation settings (Table 3). Nevertheless, the overall ratios of the single aroma compounds showed similar distributions. Therefore, probably some other aroma compounds with green odor impressions that were not quantified in this study might be responsible for the sensory perception of the aerated batch. Further studies need to be done to identify those green aroma compounds to clarify whether they are present from the original green tea infusion or synthesized by the basidiomycete. Additionally, the influence of the anti-foam agent on the metabolism should be clarified further.

## 3. Materials and Methods

### 3.1. Materials and Chemicals

Chemical compounds used in this article: dihydroactinidiolide (PubChem CID 27209), isovaleraldehyde (PubChem CID 11552), coumarin (PubChem CID 323), 2-ethyl-3,5-dimethylpyrazine (PubChem CID 26334), nonanal (PubChem CID 31289).

Acetophenone (99%), 2-ethyl-1-hexanol (99%), hexanal (99%), hexanol (98%), pentanol (98%), and 2-phenylethanol (99%) were obtained from Merck KaGaA, Darmstadt, Germany. Agar-agar (for microbiology), benzaldehyde (99%), decanoic acid (98%), limonene (95%), hexanoic acid (98%), linalool (99%), malt extract (for microbiology), octanoic acid (99%), sodium chloride (99%), *α*-terpineol (96%), and vanillin (99%) were purchased from Carl Roth GmbH & Co. KG (Karlsruhe, Germany). Antifoam 204, dihydrocarveol (95%), 2,6-dimethylpyrazine (97%), 2-ethylpyrazine (98%), geraniol (98%), *cis*-hex-3-en-1-ol (98%), 2-methoxy-4-vinylphenol (98%), 3-methyl-2(5H)-furanone (90%), nonanal (95%), octanal (99%), and terpinen-4-ol (95%) were obtained from Sigma-Aldrich (Darmstadt, Germany). γ-Butyrolactone (99%), citral (95%), coumarin (98%), decanal (96%), 2,5-dimethylpyrazine (99%), 2-ethyl-3,5(6)dimethylpyrazine (99%), eucalyptol (95%), heptanal (97%), hydroxyacetone (95%), isovaleraldehyde (98%), *cis*-jasmon (94%), methyl salicylate (98%), 6-methyl-5-hepten-2-one (98%), nonanoic acid (97%), phenylacetaldehyde (95%), and skatol (99%) were purchased from Alfa-Aesar (Kandel, Germany). Dihydroactinidiolide (95%) was obtained from Fluorochem (Hardfield, UK). Dodecanoic acid (99%), isoamylacetate (99%), isoamylalcohol (98%), and (*S*)-verbenone (94%) were obtained from Acros Organics (Geel, Belgium). Ethanol (99.9%), potassium dihydrogen phosphate (99%), and disodium phosphate (99%) were obtained from VWR Chemicals (Bruchsal, Germany). Methyl anthranilate (99%) and 2,3,5-trimethylpyrazine (98%) were purchased from J&K Scientific GmbH (Marbach/Neckar, Germany).

### 3.2. Green Tea Infusion Preparation

Whole leaf Xihu Longjing green tea (Hangzhou, Zhejiang Province, China) was used for all fermentations. For the tea infusion, 10 g/L dried tea leaves were filled into tea bags and infused for 15 min in boiled demineralized water. After removing the tea bag, the green tea was cooled down to room temperature and filtrated under sterile conditions (0.2 µm polyethersulfone (PES) membrane, GE Healthcare Europe, Freiburg, Germany). The sterile tea infusion was freshly prepared in batches prior to each fermentation experiment.

### 3.3. Fermentation of Green Tea Infusion

#### 3.3.1. Preparation of Pre-Cultures

*Mycetinis scorononius* (strain 137.83) was purchased from Westerdijk Institute Culture coollection of fungi and yeasts (Utrecht, The Netherlands). For pre-cultures, a cubic piece (1 cm^2^) of overgrown culture from malt extract (ME) agar plates (2% ME) was transferred to a culture flask (250 mL media in 500 mL flask) containing malt extract medium (2% ME) and homogenized by an Ultra-Turrax T25 homogenizer (IKA, Staufen, Germany) for 10 sec at 10,000 rpm. Pre-cultures were incubated for 7 days at 24 °C in the dark on an Innova 42R rotary shaker (Eppendorf, Hamburg, Germany) at 150 rpm. The mycelia were harvested by centrifugation (Avanti 15-R, Beckman-Coulter, Krefeld, Germany) for 10 min with 2150× *g* at 22 °C and washed 3 times with sterile water. Then, it was resuspended in 10% of the sterilized tea infusion and finally inoculated to the tea infusion.

#### 3.3.2. Fermentation in Shake Flasks

Fermentation was conducted with 250 mL green tea in a 500 mL shake flask for 64 h at 24 °C on a rotary shaker in the dark (150 rpm). After centrifugation of the cultures (10 min, 2150× *g*, 22 °C), the supernatant was transferred into pyrolyzed glass vials and sensory perception was noted by descriptive analysis (*n* = 4). Intensity was rated on a unipolar scale of 1 (very weak) to 5 (very intense). The samples were stored at −20 °C until instrumental aroma analysis.

#### 3.3.3. Bioreactor Fermentation

A laboratory-scale stirred tank bioreactor Minifors 2^®^ (Infors HT, Bottmingen, Switzerland) with embedded Linus software was used for upscaling. A glass culture vessel with a total volume of 6 L (*V_max_* = 4.5 L) and a disc stirrer with one terminal Rushton turbine impeller of 6 blades was utilized. Standard fermentation was performed in batch mode with 1 L green tea at 150 rpm and 24 °C for 64 to 72 h. During fermentation, pH and *p*O_2_ (partial dissolved oxygen) were monitored but not adjusted. At the start, *p*O_2_ measurements were set to 100% in the unfermented green tea. To analyze the effect of different fermentation parameters on the chocolate-like flavor formation, the following experiments were carried out: increase of volume (2.5 L), light protection (tank covered in aluminum foil), agitation rate (100 rpm and 250 rpm), alteration of fermentation temperature (18 °C and 30 °C), pH control (pH 5 and pH 8 using salt buffer for tea infusion preparation), and additional aeration (constant O_2_ level, 100 % and compressed air influx). The impact factors were investigated with one-factor-at-a-time approaches in analytical duplicates (Table 4).

### 3.4. Direct Immersion Stir Bar Sorptive Extraction (DI-SBSE) for Isolating Volatiles from Fermented Tea Infusion

A total of 5 mL of fermented tea infusion, 1.6 g NaCl, and 50 µL internal standard thymol (final concentration 4.16 µg/L) were added to a glass vial (20 mL). Stir bars (Twisters^®^) with a non-polar polydimethylsiloxane coating (10 mm L, 0.5 mm thick coating, Gerstel, Muehlheim an der Ruhr, Germany) were used. Extraction was performed for 2 h at room temperature and 1000 rpm. Afterwards, the twister was taken out, washed with deionized water, and dried. Then, it was inserted into a thermal desorption unit (TDU) liner (OD 6 mm, ID 4 mm, L 60 mm) (Gerstel) and desorbed in the thermal desorption unit (Gerstel). Initial heating began at 40 °C for 1 min, then rose by 120 °C/min to 220 °C, at which temperature it was held for 10 min. Volatile compounds were cryo-focused in a cooled injection system (CIS) (Gerstel) equipped with a glass wool liner (OD 3 mm, ID 2 mm, L 71 mm) with liner-in-liner principle with the TDU liner in solvent mode at 40 mL/min. After that, volatile compounds were released to a gas chromatograph starting at −100 °C for 1 min, then ramping by 12 °C/s to 230 °C, and finally held at that temperature for 5 min [7]. The samples were sniffed at gas chromatography–mass spectrometry–olfactometry (GC-MS-O) by two experienced assessors, who had participated in sensory and olfactometry training for at least 1 week at the Department of Flavor Chemistry, University of Hohenheim. GC-O analysis was performed in duplicate from two biological replicates by 3 experienced assessors (all female, mean age 25, non-smokers, *n* = 6).

### 3.5. Gas Chromatography (GC)

The 7890B gas chromatograph (GC) was combined with a 5977B mass spectrometry detector (MSD) and an OPD3 olfactory detection port (Agilent, Waldbronn, Germany and Gerstel). Gas chromatography was conducted on a J&W polar DB-WAXms column (30 m L × 0.25 mm ID × 0.25 µm film thickness) (Agilent, Waldbronn, Germany) and helium (5.0) (Westfalen, Muenster, Germany) was used as carrier gas at a constant flow rate of 1.62 mL/min. Initial oven temperature of 40 °C was held for 3 min, then rose by 5 °C/min to 240 °C, being held there for 10 min. Split ratio was set to 1:1 between the MS detector and the odor detection port (ODP) by a µFlowManager splitter (Gerstel) with a column outlet pressure of 20 kPa. The following parameters were applied for mass spectrometry: MS mode, scan mode (*m*/*z*: 40–330) at 1.562 scans/min; electron ionization energy, 70 eV; ion source temperature, 230 °C; quadrupole temperature, 150 °C; ODP 3 transfer line temperature, 250 °C; ODP mixing chamber temperature, 150 °C; ODP 3 makeup gas, N_2_ (Westfalen). Data was collected by Gerstel ODPI and Agilent Mass Hunter B07.06 in combination with Gerstel Maestro [7].

### 3.6. Compound Identification

Aroma compounds were identified by odor, retention indices (RI), comparison of mass spectra with authentic standards and published literature data, or from MS database (NIST17).

### 3.7. Semi-Quantification, Odor Activity Value (OAV), and Recombination Study

Semi-quantification of the identified aroma compounds was performed using internal standard method (IS). Response factors (*RF*) of single aroma compounds were calculated using standard compounds in water. A total of 5 mL of fermented tea infusion was added to a glass vial (20 mL), together with 1.6 g NaCl and 50 µL internal standard thymol (final concentration 4.16 µg/L). The samples were extracted for 2 h at 1000 rpm and at room temperature by DI-SBSE. Four statistical replicates were analyzed (*n* = 4). OAVs were calculated from the concentrations of the odorants determined in the fermented tea samples and the respective odor thresholds in water reported in the literature. In a recombination study, 15 assigned odorants were chosen to prepare an aroma recombinate with water as the matrix. Five odor attributes (chocolate-like, sweetish, malty, caramel-like, and woody) were defined for description of the overall aroma profile of fermented green tea and the corresponding aroma model. A unipolar five-point scale (0 to 5: 0 not detectable; 1 weak; 3 moderate; 5 strong) was used to express the odor intensities. The values were given by a sensory panel consisting of 12 experienced assessors (8 female, 4 male, all non-smokers, mean age 26).

### 3.8. Data Evaluation and Statistical Analysis

To analyze the GC-MS-O data, the software MassHunter Qualitative and Quantitative Analysis was applied. Statistical analysis was performed in Excel 2016 (Microsoft, Redmond, WA, USA) using pairwise *t*-tests (*p* ≤ 0.05). Analysis of Variance (ANOVA) with Fisher’s least significant difference (LSD) method/Tukey method as post hoc test was conducted in SPSS (IBM, Armonk, NY, USA).

## 4. Conclusions

Chocolate-like aroma flavor was formed after fermentation of green tea infusion with *M. scorodonius*, where dihydroactinidiolide, isovaleraldehyde, and coumarin out of the 15 key odorants were the most important aroma contributors to the overall aroma. The developed biotechnological process with *M. scorodonius* could be well transferred to a lab-scale bioreactor without losing the desired chocolate-like flavor impression. Alternating fermentation parameters such as medium volume, light protection, agitation rate, temperature, and pH retained the desired chocolate-like flavor impression with slight influence on the intensity. Furthermore, fermentation time was found as an important parameter that needed to be adjusted based on medium volume; while generation of a green note in aeration should be carefully monitored and controlled. The changes in the final concentrations of the 15 key aroma compounds under the various designed fermentation conditions revealed the certain concentration buffer to form the desired chocolate-like flavor. In the next steps, the fermentation broth will be extracted using food-safe solvents to prepare a concentrated aroma extract, which can then be used as a natural aroma extract in foods.

## Figures and Tables

**Figure 1 molecules-27-02503-f001:**
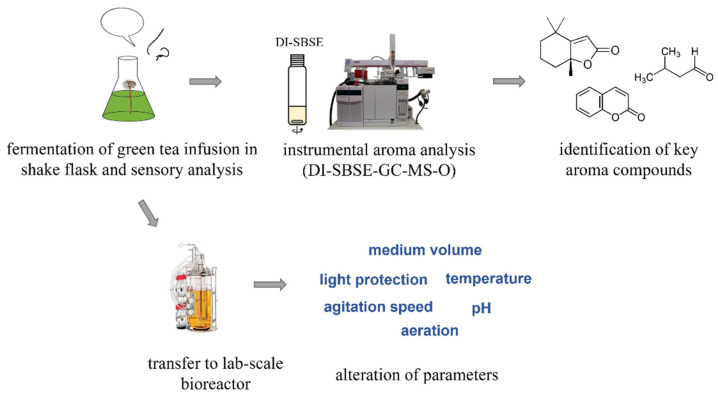
Schematic representation of the study.

**Figure 2 molecules-27-02503-f002:**
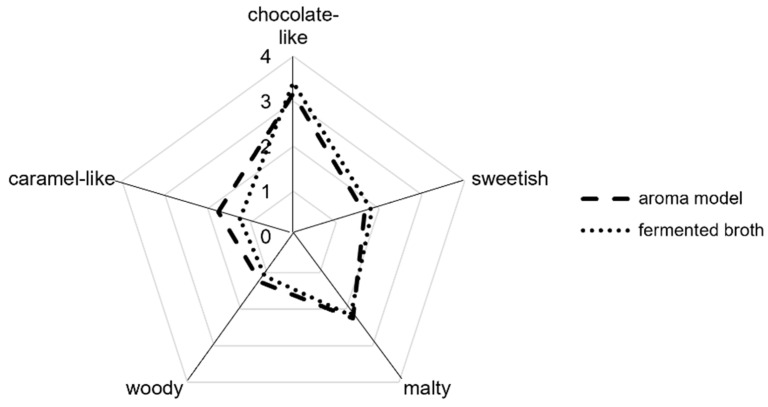
Comparative sensory analysis of the fermented green tea broth with *M. scorodonius* and the reconstituted aroma model in water by the panel (*n* = 12).

**Figure 3 molecules-27-02503-f003:**
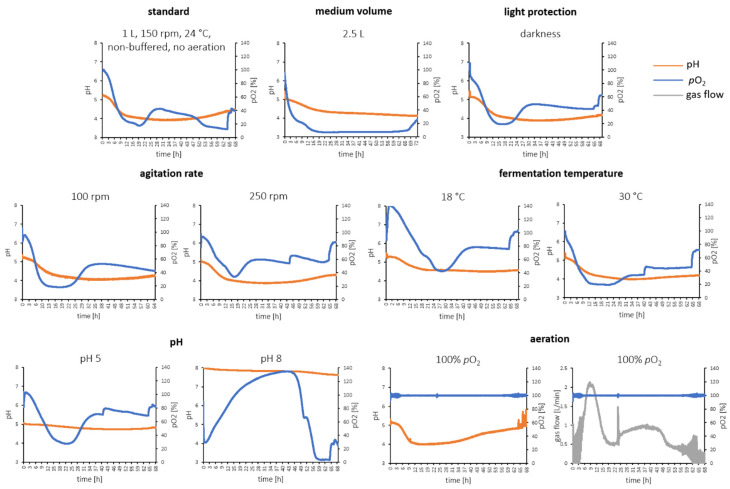
pH and *p*O_2_ monitoring in the fermentation processing with varying parameters (medium volume, light protection, agitation speed, fermentation temperature, pH, and aeration).

**Table 1 molecules-27-02503-t001:** Sensory evaluation of fermented green tea broth produced in different fermentation settings.

Set-Up		Fermentation Time (h)	Odor Impression	Odor Intensity (–)
shake flask	250 mL	64	chocolate-like, malty	3
bioreactor	1 L	68	chocolate-like, malty	2.5
	2.5 L	70	chocolate-like, woody	2
	dark	68	chocolate-like	2
	100 rpm	68	chocolate-like, licorice-like	2.5
	250 rpm	68	chocolate-like, licorice-like	2
	18 °C	68	chocolate-like	2.5
	30 °C	68	chocolate-like, malty	2
	pH 5	68	chocolate-like, sweetish	1.5
	pH 8	68	chocolate-like, sweetish, salty	2
	aerated	68	chocolate-like, green	3.5

**Table 3 molecules-27-02503-t003:** Concentrations of key aroma compounds detected in the fermented broth produced in different bioreactor settings.

	Medium Volume (L)		Light Protection	Agitation Speed (rpm)		Temperature (°C)		pH		Aeration
	1 (Standard)	2.5	Protected	100	250	18	30	5	8	100% *p*O_2_
isovaleraldehyde	7.53 ± 1.13 ^a^	15.79 ± 1.08 ^b^	10.89 ± 0.53 ^b^	3.71 ± 0.11 ^b^	7.05 ± 1.92 ^a^	43.32 ± 7.50 ^b^	22.01 ± 3.68 ^b^	6.61 ± 0.53 ^a^	2.11 ± 0.38 ^b^	68.86 ± 9.89 ^b^
octanal	1.27 ± 0.09 ^a^	0.39 ± 0.05 ^b^	1.30 ± 0.15 ^a^	2.93 ± 0.34 ^b^	0.74 ± 0.09 ^b^	3.50 ± 0.69 ^b^	3.04 ± 0.64 ^b^	0.63 ± 0.04 ^b^	0.74 ± 0.12 ^b^	23.22 ± 5.11 ^b^
(*Z*)-3-hexen-1-ol	109.62 ± 9.78 ^a^	111.95 ± 4.87 ^a^	141.71 ± 34.32 ^a^	119.98 ± 8.86 ^a^	115.13 ± 15.54 ^a^	185.47 ± 20.80 ^b^	135.14 ± 15.50 ^b^	128.17 ± 14.86 ^b^	81.83 ± 7.12 ^b^	631.31 ± 121.90 ^b^
nonanal	3.13 ± 0.35 ^a^	1.21 ± 0.17^b^	5.91 ± 0.71 ^b^	9.58 ± 1.01 ^b^	4.26 ± 0.89 ^b^	8.38 ± 0.85 ^b^	5.99 ± 2.25 ^b^	2.13 ± 0.24 ^b^	2.84 ± 0.58 ^a^	103.51 ± 13.71 ^b^
2-ethyl-3,6-dimethylpyrazine	3.72 ± 0.25 ^a^	2.79 ± 0.15 ^b^	6.94 ± 0.47 ^b^	3.79 ± 0.28 ^a^	3.57 ± 0.11 ^a^	9.12 ± 0.69 ^b^	6.66 ± 0.36 ^b^	2.93 ± 0.10 ^a^	6.62 ± 0.63 ^b^	45.88 ± 6.07 ^b^
2-ethyl-3,5-dimethylpyrazine	1.06 ± 0.11 ^a^	0.79 ± 0.04 ^b^	2.07 ± 0.14 ^b^	1.04 ± 0.21 ^a^	3.66 ± 0.14 ^b^	2.72 ± 0.21 ^b^	1.96 ± 0.14 ^b^	0.73 ± 0.07 ^b^	3.81 ± 0.11 ^b^	14.29 ± 1.87 ^b^
linalool	1.70 ± 0.16 ^a^	2.73 ± 0.11 ^b^	3.71 ± 0.09 ^b^	2.85 ± 0.02 ^b^	2.61 ± 0.55 ^b^	6.38 ± 0.26 ^b^	5.19 ± 0.33 ^b^	2.43 ± 0.34 ^b^	2.18 ± 0.09 ^b^	64.94 ± 5.96 ^b^
acetophenone	166.45 ± 8.92 ^a^	74.56 ± 1.36 ^b^	182.09 ± 44.01 ^a^	132.69 ± 6.66 ^b^	131.83 ± 4.23 ^b^	209.34 ± 22.89 ^b^	193.69 ± 60.29 ^a^	162.99 ± 29.97 ^a^	197.65 ± 22.80 ^b^	1142.48 ± 162.67 ^b^
(*S*)-verbenone	21.27 ± 1.95 ^a^	25.83 ± 1.64 ^a^	34.59 ± 2.01 ^b^	17.68 ± 0.66 ^a^	21.37 ± 3.18 ^a^	33.99 ± 6.55 ^b^	15.46 ± 1.44 ^b^	20.23 ± 1.99 ^a^	9.88 ± 0.16 ^b^	157.75 ± 14.78 ^b^
(*Z*)-geraniol	11.11 ± 0.31 ^a^	5.63 ± 0.1 ^b^	24.97 ± 0.53 ^b^	9.71 ± 0.34 ^b^	13.52 ± 1.04 ^b^	12.04 ± 0.46 ^a^	14.51 ± 0.56 ^b^	10.07 ± 0.34 ^a^	0.24 ± 0.03 ^b^	146.89 ± 10.16 ^b^
(*Z*)-carveol	40.32 ± 2.11 ^a^	5.64 ± 0.14 ^b^	26.46 ± 1.26 ^b^	26.58 ± 0.94 ^a^	16.83 ± 1.05 ^b^	17.75 ± 0.69 ^b^	30.11 ± 1.94 ^b^	10.23 ± 0.98 ^b^	5.09 ± 1.16 ^b^	418.44 ± 45.01 ^b^
(*Z*)-jasmone	2.88 ± 0.21 ^a^	2.73 ± 0.04 ^a^	7.14 ± 0.12 ^b^	3.13 ± 0.13 ^a^	3.37 ± 0.11 ^a^	8.06 ± 0.19 ^b^	4.72 ± 0.20 ^b^	3.57 ± 0.08 ^a^	3.04 ± 0.32 ^a^	33.11 ± 2.29 ^b^
2-methoxy-4-vinylphenol	67.06 ± 10.83 ^a^	43.32 ± 1.24 ^b^	104.63 ± 13.09 ^b^	56.84 ± 3.96 ^a^	57.57 ± 3.18 ^a^	120.22 ± 10.07 ^b^	101.14 ± 7.24 ^b^	58.17 ± 8.61 ^a^	53.29 ± 8.81 ^a^	679.31 ± 80.41 ^b^
dihydroactinidiolide	747.79 ± 41.01 ^a^	655.08 ± 31.09 ^b^	803.12 ± 35.57 ^b^	887.86 ± 43.84 ^b^	767.72 ± 46.08 ^a^	717.34 ± 51.46 ^b^	472.21 ± 17.74 ^b^	731.02 ± 25.07 ^a^	816.60 ± 36.45 ^b^	3384.25 ± 1187.63 ^b^
coumarin	848.01 ± 41.03 ^a^	789.27 ± 36.82 ^b^	286.88 ± 52.11 ^b^	802.07 ± 23.48 ^a^	790.03 ± 19.85 ^a^	864.43 ± 45.34 ^b^	781.81 ± 32.87 ^b^	897.96 ± 73.13 ^a^	802.48 ± 39.41 ^a^	7181.83 ± 1861.63 ^b^

^a,b^ different letters indicated a significant difference to the standard setting in bioreactor (*p* < 0.05, *n* = 4).

**Table 4 molecules-27-02503-t004:** Fermentation parameters tested in a lab-scale bioreactor.

Parameter	Standard	Variation	
medium volume	1 L	2.5 L	-
light protection	not protected from light	in the dark	-
agitation rate	150 rpm	100 rpm	250 rpm
temperature	24 °C	18 °C	30 °C
pH	non-buffered	pH 5	pH 8
aeration	no active aeration	aeration (100% *p*O_2_)	-

## Data Availability

The data presented in this study are available on request from the corresponding author. The data are not publicly available due to the large data set.

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
