# Peer review of "A Robust Fermentation Process for Natural Chocolate-like Flavor Production with *Mycetinis scorodonius"

_molecules, 2022, doi:10.3390/molecules27082503_

Round 1
Reviewer 1 Report
The authors obtained odor components with chocolate flavor by fermentation technique. The results of the study have good application prospects. However, the following aspects need further improvement before publication.
- The paper deals with very complex flavor substance preparation methods. In order to make the steps of preparation more clear to the reader. It is advisable for the authors to draw a methodological diagram of the preparation steps of the flavor substances.
- Many of the abbreviations in the paper do not give their full names when they first appear. Line 37, “EU”; line 285, “PES”; line 338, “GC-O”; “RI” and “MS” in Table 2
- The following sentences may contain errors, please check them carefully. Line 17, line 60.
- Line 99. Odor activity values (OAVs) were calculated based on the corresponding odor threshold in water. Are all the flavor components listed in Table 2 hydrophilic? Are there no hydrophobic flavor components?
- Line 350. MS mode: scan mode (m/z: 40-330) at 1.562 350 scans/min. I am very puzzled by this sentence. Do all the flavor components detected by mass spectrometry have a molecular weight less than 330?
- In Table 1, Odor Intensity is not available in units.
7. Please check the format of the following references. References 18, 20, 26.
Author Response
The authors obtained odor components with chocolate flavor by fermentation technique. The results of the study have good application prospects. However, the following aspects need further improvement before publication.
- The paper deals with very complex flavor substance preparation methods. In order to make the steps of preparation more clear to the reader, it is advisable for the authors to draw a methodological diagram of the preparation steps of the flavor substances.
- A methodology scheme was prepared and inserted as Figure 1 (line 66-68).
- Many of the abbreviations in the paper do not give their full names when they first appear. Line 37, “EU”; line 285, “PES”; line 338, “GC-O”; “RI” and “MS” in Table 2
- The abbreviations were explained when first appeared (line37, line294, line345, Table 2).
- The following sentences may contain errors, please check them carefully. Line 17, line 60.
- The sentences were checked and revised.
- Line 99. Odor activity values (OAVs) were calculated based on the corresponding odor threshold in water. Are all the flavor components listed in Table 2 hydrophilic? Are there no hydrophobic flavor components?
- Aroma compounds are known to be rather middle to non-polar than polar and therefore more hydrophobic than hydrophilic. But still, some aroma compounds are known to have more polar tendencies. In our study, since the fermentation broth is an aqueous matric, water is as an ideal reference matrix and hence the odor thresholds of the odorants in water were used to calculate their OAVs.
- Line 350. MS mode: scan mode (m/z: 40-330) at 1.562 350 scans/min. I am very puzzled by this sentence. Do all the flavor components detected by mass spectrometry have a molecular weight less than 330?
- Yes, aroma compounds are normally small molecules with an average molecular weight below 300.
- In Table 1, Odor Intensity is not available in units.
- Odor intensity was rated by smelling using an unipolar line (0-5) and therefore the intensity is unitless.
- Please check the format of the following references. References 18, 20, 26
- The references were revised. Thank you.
Reviewer 2 Report
According to this paper, the green tea infusion was fermented by M. scorodonius and formed a chocolate-like aroma. To test the robustness of the fermentation process, the effects of different fermentation parameters on the biosynthesis of chocolate-like flavor were tested in a laboratory-scale bioreactor. It is a topic of interest to the researchers in the related areas but the paper needs improvement before acceptance for publication. My detailed comments are as follows:
- line81-83: What do these enzymes have to do with natural flavor or the resulting chocolate-like aroma?Why no further explanation?
- line162, line213: What exactly does“three broths”and “three samples” mean?
- Please explain the phenomenon of dissolved oxygen and whether it affects the experimental results?
- Please combine the contents of Table 2 and Table 3, and supplement the Odor Threshold and OAV of other compounds.
- In the recombination study, 15 odorants assigned were chosen to prepare an aroma recombinate with water as matrix. Would the flavor profile be affected by the other compounds (although the OAVs < 1)? Andtheinteractions between the compounds should be considered in the recombination study?
- 6. There are lots of minor mistakes in this manuscript, should be carefully modified by authors, such as:
line60: pleasant, “p” should be capitalized.
line351: ion source temperature.: 230℃, wrong punctuation.
Author Response
According to this paper, the green tea infusion was fermented by M. scorodonius and formed a chocolate-like aroma. To test the robustness of the fermentation process, the effects of different fermentation parameters on the biosynthesis of chocolate-like flavor were tested in a laboratory-scale bioreactor. It is a topic of interest to the researchers in the related areas but the paper needs improvement before acceptance for publication. My detailed comments are as follows:
- line81-83: What do these enzymes have to do with natural flavor or the resulting chocolate-like aroma? Why no further explanation?
- Some more information on role of the enzymes were given (line 85-88).
- line162, line213: What exactly does “three broths” and “three samples” mean?
- The description was specified (line167-169; line220-222).
- Please explain the phenomenon of dissolved oxygen and whether it affects the experimental results?
- It reported in literature that high proportion of dissolved oxygen can boost the growing and the metabolism of various microorganisms, since there is a correlation between amount of produced energy and available oxygen. Therefore, it’s expected that a higher dissolved oxygen proportion will change the secretom of the fungus and therefore alternates the final sensory perception as shown in this study (line 245-264)
- Please combine the contents of Table 2 and Table 3 and supplement the Odor Threshold and OAV of other compounds.
- The tables were combined and additional information was inserted (Table 2).
- In the recombination study, 15 odorants assigned were chosen to prepare an aroma recombinate with water as matrix. Would the flavor profile be affected by the other compounds (although the OAVs < 1)? And the interactions between the compounds should be considered in the recombination study?
- OAV below 1 suggests a minor contribution of the compound for the overall aroma if there is no synergistic effect. The OAV concept is a good approach to estimate the contribution of aroma active compounds to the overall aroma. A disadvantage of this concept is the absence of synergistic or antagonistic interactions between compounds among each other with the matrix. Nevertheless, in our study we could prove that our recombinate with 15 determined key aroma compounds (OAVs ≥ 1) could well recreate the chocolate-like and malty odor impression of the fermented broth.
- There are lots of minor mistakes in this manuscript, should be carefully modified by authors, such as line60: pleasant, “p” should be capitalized/ line351: ion source temperature.: 230℃, wrong punctuation
- The manuscript was revised and modified (line 60, line 359). Thank you.
Reviewer 3 Report
Line 2: This article describes the bioconversion of a green tea extract into a chocolate-like flavor preparation. The title should be changed to reflect the nature of the process i.e. bioconversion
The use of the organism is interesting and the GC-O work has merit.
Lines 159 - 254: However, given the lack of impact on the bioconversion of the different parameters in section 2.2, and the lack of physiological data on the state of growth of the organism, this section should be significantly reduced or moved to supplemental data in its entirety. The poor quality of the write up of this section also is an issue and uses non scientific language, such a "huge".
Methods
Line 303: Please add the number of panelists used for the descriptive analysis work
Line282 Please explain what for the tea was in before extraction - whole leaf or powdered?
Line 299: change "fermentation" to bioconversion
Author Response
- Line 2: This article describes the bioconversion of a green tea extract into a chocolate-like flavor preparation. The title should be changed to reflect the nature of the process i.e. bioconversion
- Both definitions are quite similar and can be used almost as synonyms. Whereas bioconversion describes the usage of energetically low and unusable biological precursors (biomass) into usable products, fermentations are biochemical reactions for secondary metabolites from different sources. Since in our study no energetically low or unusable substrate was used, we prefer sticking to the term “fermentation”.
- The use of the organism is interesting, and the GC-O work has merit.
- We appreciate the comment.
- Lines 159 - 254: However, given the lack of impact on the bioconversion of the different parameters in section 2.2, and the lack of physiological data on the state of growth of the organism, this section should be significantly reduced or moved to supplemental data in its entirety. The poor quality of the write up of this section also is an issue and uses non-scientific language, such a "huge".
- The section was revised carefully. Section 2.2 is important to show the robustness of our developed process (line150-264) and therefore was kept at this point.
Methods:
- Line 303: Please add the number of panelists used for the descriptive analysis work
- The number of panelists was added (line311).
- Line282 Please explain what for the tea was in before extraction - whole leaf or powdered?
- More information was added (line290).
- Line 299: change "fermentation" to bioconversion
- Please see comment above.
We appreciate the valuable reviewing process and hope to have answered all of the requests satisfactorily.
Round 2
Reviewer 1 Report
The previously mentioned issues have been largely resolved and the editor may consider accepting the manuscript.
Reviewer 2 Report
it's ok to public after revision.